# Composition and diversity of meibum microbiota in meibomian gland dysfunction and the correlation with tear cytokine levels

Ubonwan Rasaruck[1], Ngamjit Kasetsuwan[1,2]*, Thanachaporn Kittipibul[1,2], Pisut Pongchaikul[3,4,5], Tanittha Chatsuwan[6,7]

1 Department of Ophthalmology, Faculty of Medicine, Chulalongkorn University, Bangkok, Thailand, 2 Excellence Center of Cornea and Limbal Stem Cell Transplantation, Department of Ophthalmology, King Chulalongkorn Memorial Hospital and Faculty of Medicine, Chulalongkorn University, Bangkok, Thailand, 3 Chakri Naruebodindra Medical Institute, Faculty of Medicine Ramathibodi Hospital Mahidol University, Samut Prakarn, Thailand, 4 Integrative Computational Bioscience (ICBS) Center, Mahidol University, Nakorn Pathom, Thailand, 5 Institute of Infection, Veterinary and Ecological Sciences, University of Liverpool, Liverpool, United Kingdom, 6 Department of Microbiology, Faculty of Medicine, Chulalongkorn University, Bangkok, Thailand, 7 Center of Excellence in Antimicrobial Resistance and Stewardship, Faculty of Medicine, Chulalongkorn University, Bangkok, Thailand

* ngamjitk@gmail.com

**Data Availability Statement:** The datasets generated and analyzed during the current study are available in the NCBI short read archive repository under BioProject accession number

## Abstract

Meibomian gland dysfunction (MGD) leads to meibum stasis and pathogenic bacteria proliferation. We determined meibum microbiota via next-generation sequencing (NGS) and examined their association with tear cytokine levels in patients with MGD. This cross-sectional study included 44 moderate–severe patients with MGD and 44 healthy controls (HCs). All volunteers underwent assessment with the ocular surface disease index questionnaire, Schirmer without anesthesia, tear break-up time, Oxford grading of ocular surface staining, and lid and meibum features. Sample collection included tears for cytokine detection and meibum for 16S rRNA NGS. No significant differences were observed in the α-diversity of patients with MGD compared with that in HCs. However, Simpson's index showed significantly decreased α-diversity for severe MGD than for moderate MGD (p = 0.045). Principal coordinate analysis showed no significant differences in β-diversity in meibum samples from patients with MGD and HCs. Patients with MGD had significantly higher relative abundances of *Bacteroides* (8.54% vs. 6.00%, p = 0.015) and *Novosphingobium* (0.14% vs. 0.004%, p = 0.012) than the HCs. Significantly higher interleukin (IL)-17A was detected in the MGD group than in the HC group, particularly for severe MGD (p = 0.008). Although *Bacteroides* was more abundant in the MGD group than in the HC group, it was not positively correlated with IL-17A. The relationship between core meibum microbiota and tear cytokine levels remains unclear. However, increased *Bacteroides* and *Novosphingobium* abundance may be critical in MGD pathophysiology.

PRJNA894103 (https://www.ncbi.nlm.nih.gov/bioproject/PRJNA894103?fbclid=IwAR1fCQz3ucOXzBvuA7GB0_xPiLaE5BKUCzBn5jWzMogi1dcIsmdFZuZUKNk).

**Funding:** This research was supported by the Ratchadapisek Sompoch Endowment Fund, Faculty of Medicine, Chulalongkorn University (GA64/020) (U.R.); 90th Anniversary of Chulalongkorn University Scholarship, Chulalongkorn University (N.K.); and Research Grant of The Medical Council of Thailand (N.K.). Ubonwan Rasaruck was supported under H.M. by the King Bhumibol Adulyadej's 72nd Birthday Anniversary Scholarship for the Master's Degree program, Chulalongkorn University. The funder had no role in study design, data collection and analysis, decision to publish, or preparation of the manuscript.

**Competing interests:** The authors have declared that no competing interests exist.

## Introduction

Meibomian gland dysfunction (MGD) is a multifactorial disorder that may cause eyelid and conjunctival inflammation, corneal damage, tear film instability, and changes in microbial composition [1]. Common symptoms of MGD include visual disturbances with ocular discomfort, itching, and photophobia. Inflammatory obstructive MGD highly correlates with hordeolum, conjunctivitis, and keratitis [2]. The prevalence of MGD is 4–20% in Caucasians and 46–69% in the Asian population [3]. Increased tear evaporation from MGD is the most common form of dry eye disease (DED) [4].

One of the key pathophysiologies of MGD is hyperviscous meibum, which leads to meibum stasis and eventually results in the proliferation of pathogenic bacteria on the ocular surface as well as inside the meibomian gland [5]. Alterations to bacterial composition on the ocular surface consequently aggravate the production of inflammatory cytokines, resulting in subclinical inflammation and epithelial hyperkeratinization [5].

Owing to the proliferation of certain pathogenic bacteria, the diversity of meibum microbiota in patients with MGD is expected to decrease compared with that in healthy individuals. Increased inflammatory products from specific types of bacteria are related to ocular irritation symptoms in patients with MGD [2] and may be associated with MGD severity [6]. Meibomian gland obstruction elevates intraglandular pressure and stimulates mitogen-activated protein kinase activity, producing inflammatory cytokines. Accordingly, tear inflammatory cytokines are potential biomarkers of MGD [5]. Previous studies have shown that increased cytokine levels were detected in moderate to severe cases of DED [6–11]. The specific tear inflammatory cytokines correlated with clinical parameters of patients with MGD are interleukin (IL)-6, IL-17A, and IL-1β [6, 12, 13].

The microbiota in the ocular surface differs considerably from that in regions such as the skin, eyelid margin, and conjunctiva [14]. Next-generation sequencing (NGS) offers targeted metagenomic sequencing with a higher rate of bacterial detection than that of conventional culture-based tests [15]. In this study, we aimed to determine the meibum microbiota by NGS and its association with tear cytokine levels (IL-6, IL-17A, and IL-1β) in patients with MGD compared with healthy controls (HCs).

## Materials and methods

In this study, 44 patients with MGD between 40 and 80 years of age and 44 age- and sex-matched HCs were enrolled in the outpatient clinic of the Department of Ophthalmology, King Chulalongkorn Memorial Hospital, Bangkok, Thailand, from 6 November 2021 to 8 April 2022. The sample size of participants in each group was calculated by comparison of two independent means and standard deviation from a previous study [1]. The study was conducted in accordance with the stipulations of the Declaration of Helsinki, approved by the Institutional Review Board, Faculty of Medicine, Chulalongkorn University (IRB No. 289/64 and COA No. 920/2021), and was registered in the Thai Clinical Trial Registration (TCTR No. TCTR20210221002). Written informed consent was obtained from all participants. Inclusion criteria for the MGD group included: (1) ocular surface disease index (OSDI) score ≥ 13; (2) tear break-up time (TBUT) < 10 s or ocular surface staining by fluorescein consistent with the diagnosis of dry eye (> 5 corneal spots, > 9 conjunctival spots, or lid margin staining of ≥ 2 mm length and ≥ 25% width) [16]; (3) Schirmer test without anesthesia ≥ 5 mm/5 min; and (4) diagnosis of moderate to severe MGD (MGD stage 3–4) in accordance with the International Workshop on Meibomian Gland Dysfunction [17, 18]. The patients were diagnosed as having moderate MGD if they had at least one of the following criteria: (1) OSDI score 23–32; (2) lid margin vascularity or plugged meibomian orifice; (3) meibum quality score 8–12; (4) meibum

expressibility grade 2; (5) Oxford grading score 4–10. The patients were diagnosed as having severe MGD if they had at least one of the following criteria: (1) OSDI score 33–100; (2) displacement of the mucocutaneous junction; (3) meibum quality score $\geq$ 13; (4) meibum expressibility grade 3; (5) Oxford grading score 11–15. Meibum quality is assessed at eight glands in the central third of the lower eyelid; grade 0 = clear, grade 1 = cloudy, grade 2 = cloudy with granular debris, grade 3 = thick, like toothpaste. Meibum expressibility is assessed at five glands in central third of lower eyelid; grade 0 = all, grade 1 = 3–4, grade 2 = 1–2, grade 3 = 0, modified from Geerling et al. [18] (S1 File). Inclusion criteria for the HCs included: (1) OSDI score < 13; (2) no history of dry eye; (3) TBUT $\geq$ 10 s; (4) ocular surface staining by fluorescence not consistent with the diagnosis of dry eye; and (5) lid and meibomian gland evaluation not compatible with the diagnosis of MGD. All inclusion criteria should have been present in participants. If both eyes met the inclusion criteria, the investigator randomly selected only one eye. We excluded volunteers based on the following: (1) history of topical or systemic antibiotics or topical ophthalmic drops (except for preservative-free artificial tears) within the past 3 months; (2) history of contact lens wear within the last 3 months; (3) ongoing ocular allergy, infection, or inflammation irrelevant to dry eye or MGD; (4) history of ocular surgery within the last 6 months; (5) history of systemic conditions that affect the ocular surface; and (6) healthcare workers, pregnant women, or persons with mental illness. The participants were appointed for data collection which was performed in the following order in the same visit: (1) The baseline characteristics and OSDI questionnaire were recorded; (2) Schirmer test without anesthesia and tear sample collection were performed; (3) The ocular surface, lid, and meibomian gland assessments were performed by Ubonwan Rasaruck (U.R.) following the MGD severity grading scale, modified from Geerling et al. [18] (S1 File); (4) The meibum samples were collected by U.R.

Tear samples were collected using a Schirmer strip without anesthesia based on a sterile technique [19]. Strips were stored in a 2-mL centrifuge tube, kept in an ice-filled container for 20–60 min, and stored at −80 ˚C until cytokine analysis. The meibum samples were collected using a sterile glove. The eyelid margin was sterilized with 10% povidone-iodine, cleaned with sterile saline, and wiped with a dry cotton swab. The meibum was squeezed using a meibomian gland compressor and collected with a dry sterile cotton swab. The swab was placed into a DNase-free tube with DNA/RNA shield solution (Zymo, Irvine, CA, USA) and stored at −20 ˚C for later analysis. The blank swab was collected on the same day of sample collection by exposing the sterile swab in the atmosphere of the investigation room for 10 sec and placing the swab in a DNase-free tube with DNA/RNA shield solution. The blank swab was processed in an identical manner to the test samples and sequenced with the test samples. DNA was extracted from the meibum samples using the QIAmp® DNA Microbiome Kit (QIAGEN, Hilden, Germany). The concentration of extracted DNA was measured in ng/ul.

The specific tear cytokines, namely IL-6, IL-17A, and IL-1β, were extracted from the Schirmer strips. The assay buffer contained 200 µL of 1% BSA in phosphate-buffered saline (PBS) with the addition of sodium azide for preservation and was added to each centrifuge tube. Subsequently, the tube was incubated in a rocker at 25 ˚C for 3 h and kept in an ice-filled container. Each Schirmer strip was transferred to a new centrifuge tube with the strip placed at the 25-mm mark on the sealed tube cap and centrifuged at $100 \times g$ for 10 s. Subsequently, the contents of the new tube was mixed with those of the old tube containing residual tears and buffer. The concentrations of tear cytokines (pg/mL) were analyzed by Luminex using a cytokine kit (Bio-Plex Pro™; Bio-Rad, Hercules, CA, USA) [19].

The NGS analysis was conducted at Omics Sciences and Bioinformatics Center, Chulalongkorn University, Bangkok, Thailand. After library preparation, 16S rRNA gene amplification was performed using 341F and 805R primers targeting V3–V4 variable regions and the 2 sparQ HiFi PCR Master Mix (QuantaBio, Beverly, MA, USA). The contaminant filtering step

was performed by using positive and negative controls and filtering out rare amplicon sequence variants. The amplification process started with a denaturation step at 98 ˚C for 2 min; 30 cycles at 98 ˚C for 20 s, 55 ˚C for 30 s, and 72 ˚C for 1 min; and a final extension step at 72 ˚C for 1 min. Subsequently, the 16S amplicon was purified by sparQ Puremag Beads (QuantaBio), labeled with 2.5 μL of each Nextera XT index primer in a 25 μL polymerase chain reaction (PCR), and underwent ten cycles of PCR as mentioned above. The final PCR products were purified, pooled, and diluted to a final loading concentration of 4 pM. Lastly, an Illumina MiSeq (Illumina, San Diego, CA, USA) was used to perform cluster generation and 250 bp paired-end read sequencing.

Paired-end reads were qualified and quantified using FastQC software (version 0.11.8, Babraham Institute, Cambridge, UK). Chimeric sequences were removed using the VSearch software (version 2.21, http://drive5.com/usearch/). Operational taxonomic units (OTUs) were assigned and used to perform alpha diversity, beta diversity, and differential abundance analyses. Alpha diversity, including the rarefaction curve, abundance-based coverage estimator (ACE), Chao1, Pielou, Shannon, and Simpson indices, as well as beta diversity were calculated using the MicrobiotaProcess package in R (version 1.8.2; R Foundation for Statistical Computing, Vienna, Austria). Differential abundance was analyzed using a negative binomial distribution-based model in the DESeq2 package, implemented in R. Briefly, the tables containing OTUs, taxa, and meta-tables were combined and converted into phyloseq format, which is the standard format for microbiome analysis. Analyses were conducted using the DESeq2 function with an adjusted p-value of 0.01.

All statistical analyses were performed using R (version 1.8.2; R Foundation for Statistical Computing, Vienna, Austria). Baseline characteristics were analyzed using descriptive statistics. Student's t-test was used to compare the OSDI score, TBUT, Schirmer test, Oxford score, and meibum quality between the MGD and HC groups. The Mann–Whitney U test was used to compare the α-diversity index and relative abundance between the two groups. One-way analysis of variance (ANOVA) was used to compare the OSDI score, TBUT, Schirmer test, Oxford score, meibum quality, α-diversity index, the relative abundance of meibum microbiota, and tear cytokine levels among the severe MGD, moderate MGD, and HC groups. Fisher's exact test was used for comparing lid margin irregularities and plugged MG orifices among the three groups. Chi-square tests were used for comparing lid margin vascularity and displacement of mucocutaneous junction (MCJ) among the three groups. Spearman correlation was performed to determine the association between core meibum microbiota and tear cytokine levels. Logistic regression was used to compare the relative abundance of meibum microbiota and lid assessment. Pearson correlation coefficient was used for comparing the relative abundance of meibum microbiota and TBUT, Schirmer test, OSDI, Oxford grading of ocular surface staining, and meibum quality. Statistical significance was set at $p < 0.05$.

## Results

### Clinical characteristics

We collected 88 meibum samples and 88 tear samples from 44 eyes of 44 patients with moderate to severe MGD and 44 eyes of 44 age- and sex-matched HCs. We enrolled 29 males (33%) and 59 females (67%) with a mean age of 61.0 years in the MGD group and 59.8 years in the HC group. Demographic data of the study participants are shown in Table 1.

### Alpha diversity

The rarefaction curves of MGD and HCs ended at saturation platforms, demonstrating an appropriate sequencing data size. The α-diversity was compared between the MGD and HC

**Table 1. Demographic data and clinical characteristics of patients with meibomian gland dysfunction and healthy controls.**

| | MGD | Severe MGD | Moderate MGD | Healthy controls | p-value (MGD vs. HC) | p-value (severe MGD vs. moderate MGD vs. HC) |
|---|---|---|---|---|---|---|
| Number of participants | 44 | 26 | 18 | 44 | | |
| Age (mean ± SD) | 61.0 ± 9.8 | 61.6 ± 9.9 | 60.1 ± 9.9 | 59.8 ± 10.3 | | |
| Sex ratio (female: male) | 29:15 | 12:14 | 17:1 | 30:14 | | |
| Laterality | | | | | | |
| • Left eyes (number) | 22 | 11 | 11 | 20 | | |
| • Right eyes (number) | 22 | 15 | 7 | 24 | | |
| OSDI (Mean ± SD) | 26.9 ± 15.0 | 32.9 ± 16.9 | 18.3 ± 3.5 | 6.8 ± 4.6 | < 0.001 | <0.001* |
| TBUT (Mean ± SD) | 4.8 ± 1.1 | 4.7 ± 1.2 | 4.9 ± 1.1 | > 10 | < 0.001 | < 0.001† |
| Schirmer test (Mean ± SD) | 10.0 ± 7.3 | 10.1 ± 8.6 | 9.6 ± 4.5 | 15.3 ± 10.6 | 0.006 | 0.022 |
| Oxford (Mean ± SD) | 3.7 ± 1.6 | 3.9 ± 1.5 | 3.4 ± 1.9 | 0.3 ± 0.7 | < 0.001 | < 0.001† |
| Lid assessment | | | | | | |
| • Lid margin irregularities (present, number) | 15 | 12 | 3 | 0 | < 0.001 | < 0.001† |
| • Lid margin vascularity (present, number) | 45 | 27 | 18 | 5 | < 0.001 | < 0.001† |
| • Plugged MG orifices (present, number) | 11 | 7 | 4 | 0 | 0.002 | 0.002† |
| • Displacement of MCJ (present, number) | 18 | 18 | 0 | 0 | < 0.001 | < 0.001‡ |
| Meibum quality (Mean ± SD) | 15.1 ± 5.7 | 18.3 ± 5.2 | 10.3 ± 1.7 | 0 | < 0.001 | < 0.001 |

Abbreviations: HC, healthy controls; MGD, meibomian gland dysfunction; OSDI, ocular surface disease index; Oxford, Oxford grading scale for corneal staining; TBUT, tear film break-up time

*Comparison between moderate MGD vs. HC, severe MGD vs. HC, and moderate MGD vs. severe MGD

†Comparison between moderate MGD vs. HC and severe MGD vs. HC

‡Comparison between moderate MGD vs. severe MGD and severe MGD vs. HC

groups; however, no significant differences were observed (p > 0.05; Fig 1). Simpson's index showed significantly decreased α-diversity in patients with severe MGD than those with moderate MGD. (p = 0.045; Fig 1). However, there was no difference in α-diversity between patients with severe MGD and HCs and patients with moderate MGD and HCs (p > 0.05; Fig 1).

## Beta diversity

The Bray–Curtis dissimilarity indices, Jaccard's distance, weighted UniFrac distance, and unweighted UniFrac distance were used to compare the differences in the entire taxonomic composition. Principal coordinate analysis (PCoA) showed no clear distinction between the meibum samples from the MGD and HC groups (p > 0.05; permutational multivariate ANOVA [PERMANOVA]). For the severe MGD, moderate MGD, and HC subgroups, there was no statistically significant difference in the beta diversity index between each subgroup (p > 0.05; PERMANOVA).

## Taxonomic composition of meibum microbiota

We detected 31 phyla in the samples. The predominant bacteria in both MGD and HCs were Firmicutes (46.73% vs. 45.99%), Actinobacteria (20.15% vs. 21.40%), Proteobacteria (17.26%

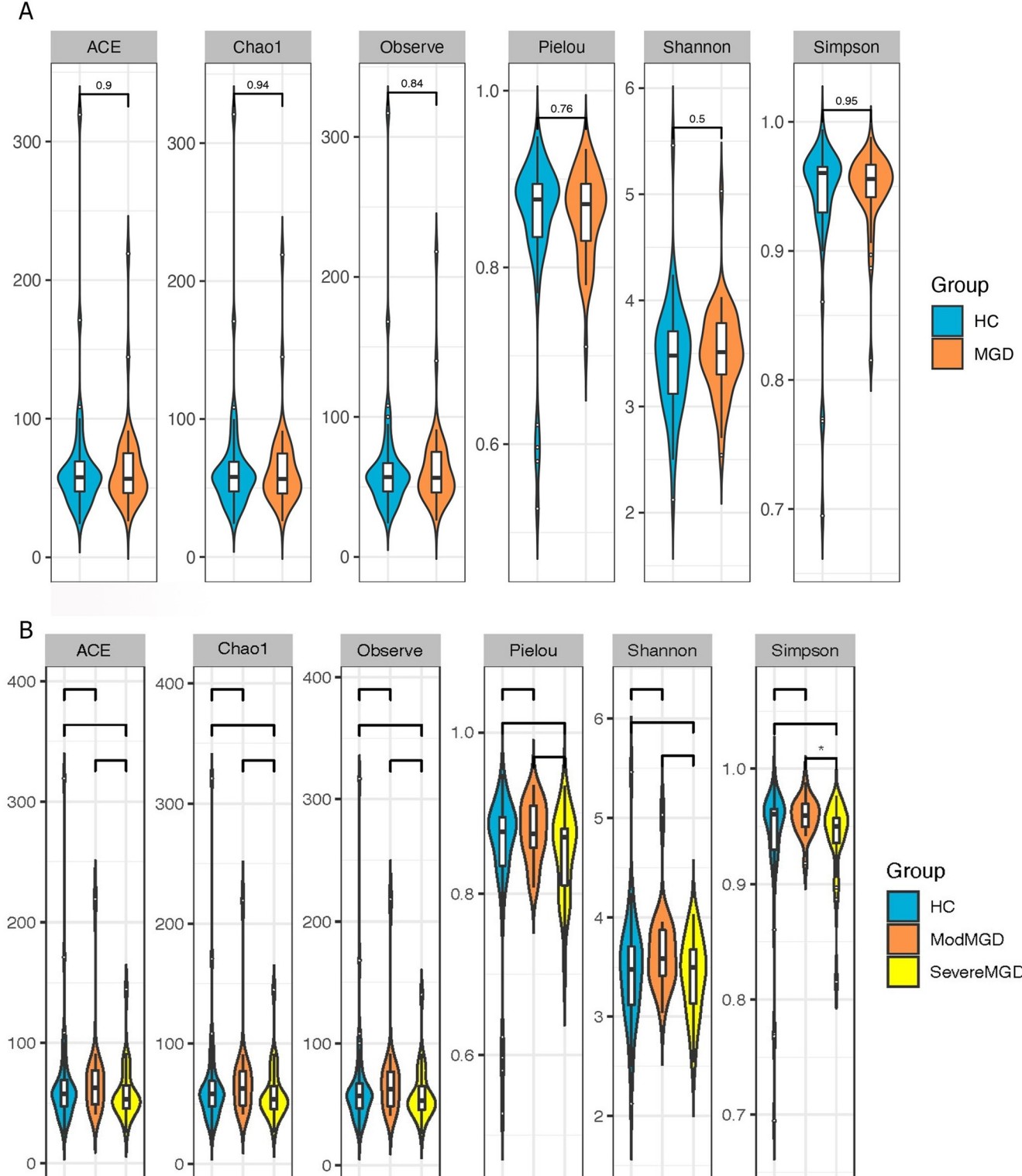

**Fig 1. This is the schematic showing α-diversity between groups based on ACE, Chao1, Observe, Pielou, Shannon, and Simpson's index: (A) between HC and MGD groups, (B) between HC, moderate MGD, and severe MGD groups (*p = 0.045).**

vs. 18.10%), and Bacteroidetes (12.46% vs. 10.47%). At the genus level, samples from the MGD group had significantly higher relative abundances of *Bacteroides* (8.54% vs. 6.00%, p = 0.015)) [Fig 2; Table 2] than those from the HC group. Subgroup analysis showed that patients with severe MGD had a significantly higher relative abundance of *Bacteroides* than the HCs (9.00%

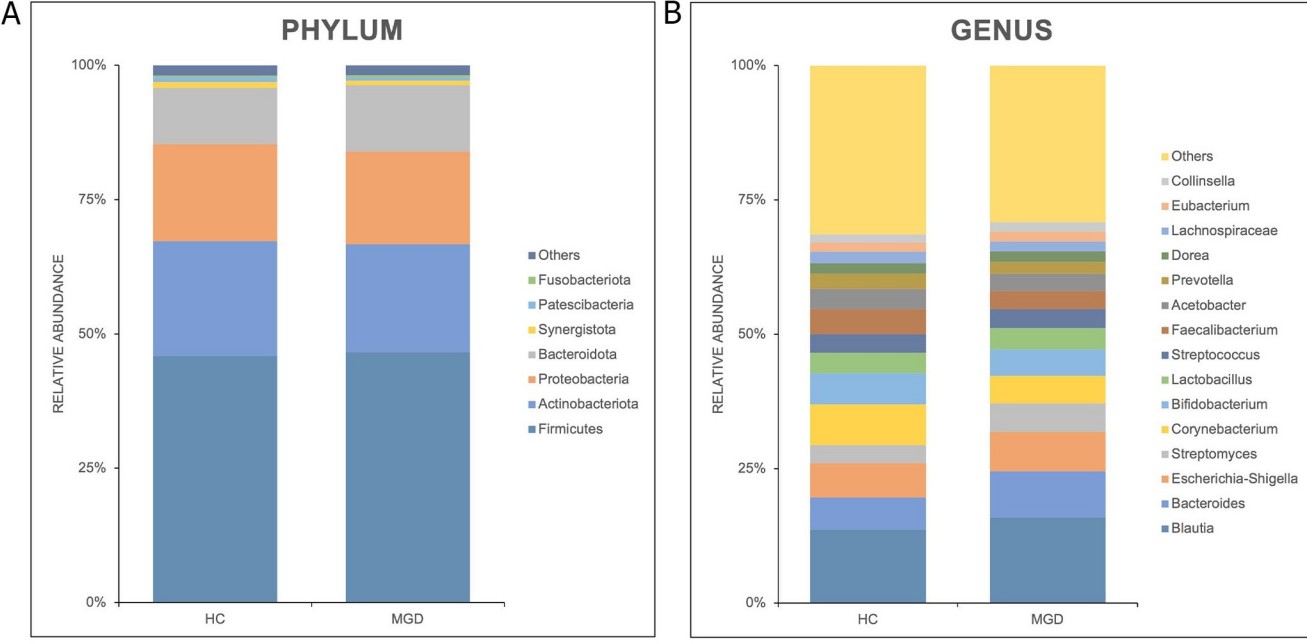

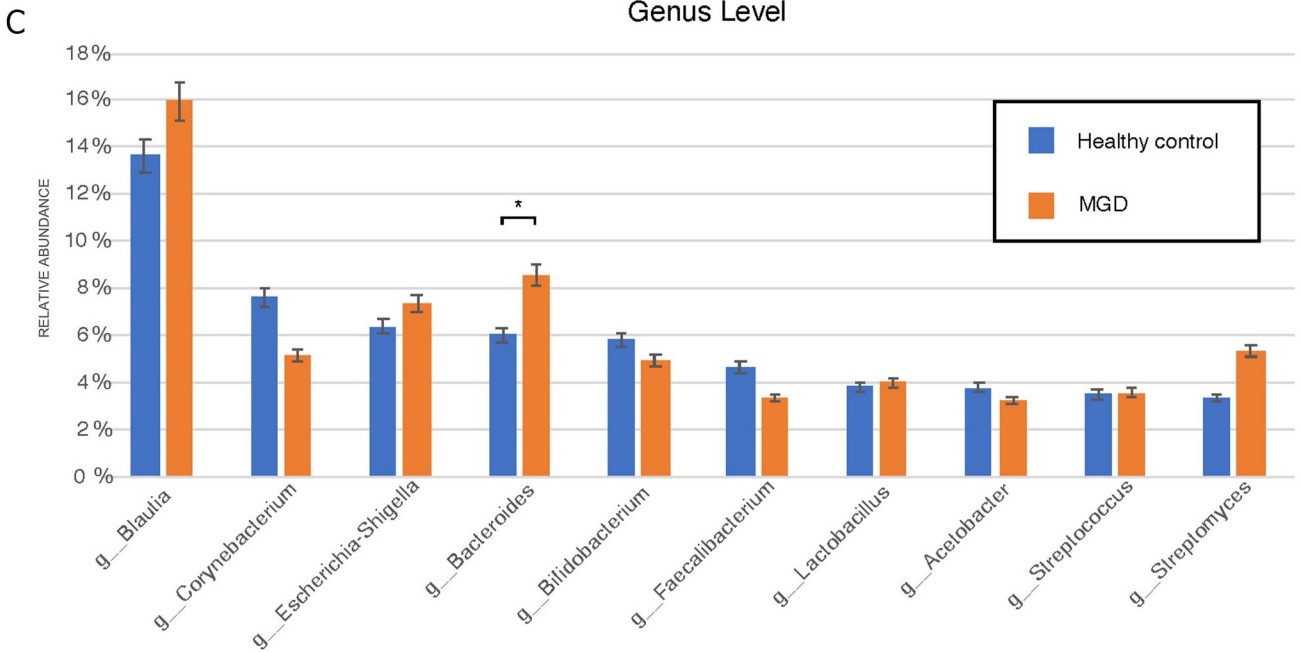

**Fig 2.** This is the relative abundance of meibum microbiota between MGD and HC groups at (A) phylum and (B) genus levels. (C) Statistically significantly higher relative abundance of *Bacteroides* in the MGD group than in the HC group (8.54% vs. 6.00%, p = 0.015). Abbreviations: p, phylum; g, genus; HC, healthy controls; MGD, meibomian gland dysfunction.

**Table 2. Relative abundance of predominant meibum microbiota.**

| | MGD (n = 44) | Severe MGD (n = 26) | Moderate MGD (n = 18) | HCs (n = 44) | p-value (HCs vs. MGD) | p-value (3 groups) |
|---|---|---|---|---|---|---|
| p__Firmicutes | 46.75 ± 12.63% | 46.59 ± 13.02% | 46.98 ± 12.41% | 45.99 ± 15.16% | 0.799 | 0.964 |
| p__Actinobacteria | 20.15 ± 11.48% | 21.17 ± 13.35% | 18.67 ± 8.22% | 21.40 ± 13.36% | 0.638 | 0.582 |
| p__Proteobacteria | 17.26 ± 10.29% | 17.13 ± 11.38% | 17.44 ± 8.80% | 18.09 ± 12.59% | 0.732 | 0.943 |
| p__Bacteroidota | 12.46 ± 6.13% | 12.78 ± 6.14% | 11.99 ± 6.27% | 10.47 ± 6.00% | 0.127 | 0.297 |
| g__Blautia | 15.95 ± 7.62% | 15.49 ± 7.64% | 16.62 ± 7.76% | 13.65 ± 8.36% | 0.181 | 0.377 |
| g__Bacteroides | 8.54 ± 5.17% | 9.00 ± 5.02% | 7.86 ± 5.44% | 6.00 ± 4.42% | **0.015** | **0.045** |
| g__Escherichia-Shigella | 7.35 ± 4.24% | 7.76 ± 4.68% | 6.74 ± 3.54% | 6.38 ± 4.42% | 0.299 | 0.483 |
| g__Streptomyces | 5.30 ± 5.54% | 4.30 ± 4.49% | 6.74 ± 6.67% | 3.34 ± 4.04% | 0.614 | 0.134 |
| g__Corynebacterium | 5.14 ± 7.17% | 6.47 ± 8.56% | 3.23 ± 3.97% | 7.59 ± 13.33% | 0.286 | 0.072 |
| g__Bifidobacterium | 4.89 ± 3.47% | 4.95 ± 3.71% | 4.79 ± 3.19% | 5.82 ± 4.96% | 0.286 | 0.588 |
| g__Lactobacillus | 3.98 ± 4.65% | 2.71 ± 3.52% | 5.83 ± 5.51% | 3.79 ± 4.73% | 0.841 | 0.113 |
| g__Streptococcus | 3.55 ± 3.99% | 4.60 ± 4.61% | 2.04 ± 2.04% | 3.47 ± 3.79% | 0.930 | 0.029 |
| g__Faecalibacterium | 3.35 ± 2.96% | 3.76 ± 3.19% | 2.75 ± 2.57% | 4.65 ± 5.19% | 0.154 | 0.162 |
| g__Acetobacter | 3.20 ± 5.17% | 2.46 ± 4.40% | 4.27 ± 6.09% | 3.74 ± 5.20% | 0.623 | 0.435 |

Abbreviations: p, phylum; g, genus: HC, healthy controls; MGD, meibomian gland dysfunction

vs. 5.99%, p = 0.045). *Novosphingobium*, although not among the ten most predominant microbiota, had significantly higher relative abundances in the MGD group, compared with the HCs. (0.14% vs. 0.004%, p = 0.012) Nevertheless, the post-hoc test revealed no significant differences in the relative abundance of *Novosphingobium* among the three groups (p > 0.05).

## Core meibum microbiota

The abundance–occurrence method was used to identify the core meibum microbiota, with the minimal relative abundance threshold set at 0.001% and the occurrence cut-off point set at 50% [20]. The core meibum microbiota at the genus level in both the MGD and HC groups were the *Blautia, Bacteroides, Escherichia-Shigella, Streptomyces, Corynebacterium, Bifidobacterium, Lactobacillus, Streptococcus, Faecalibacterium, Acetobacter, Prevotella, Lachnospiraceae, Dorea, Collinsella, Staphylococcus, Pseudomonas*, and *Eubacterium hallii* groups.

## Differential abundance analysis

The *Bacteroides* were significantly more abundant (p < 0.001), and *Nocardia* and *Obscuribacteraceae* were significantly less abundant in the MGD group than in the HC group (p < 0.001) [Fig 3].

## Correlation between core meibum microbiota and tear cytokine levels

IL-17A levels were significantly higher in the MGD group than in the HC group (p = 0.008). Subgroup analysis also demonstrated that patients with severe MGD had significantly higher IL-17A levels than those in the HCs (p = 0.02). IL-6 levels increased in the MGD group, compared with the HC group; however, this increase was not statistically significant (Table 3). Although there were significantly higher levels of IL-17A in MGD, IL-17A was not correlated with the relative abundance of *Bacteroides*; however, a moderate negative correlation with IL-1β was observed (Spearman's rho = −0.31, p < 0.05) [Fig 4].

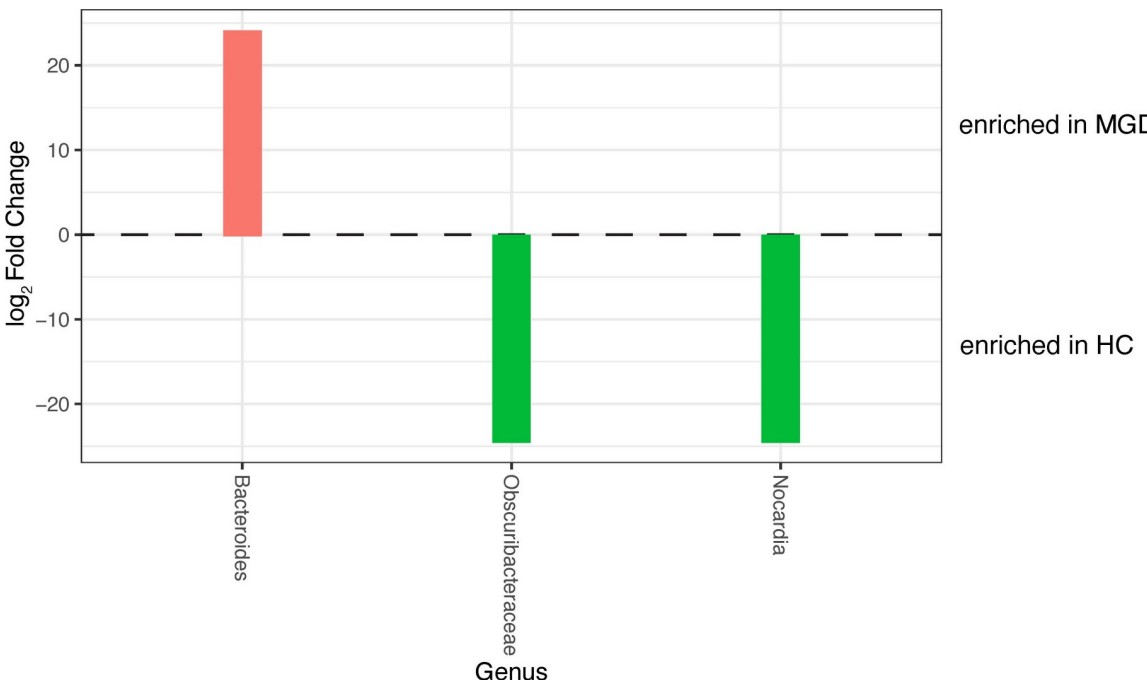

**Fig 3. This is the differential abundance analysis.** The figure demonstrates that *Bacteroides* was enriched in MGD, whereas *Nocardia* and *Obscuribacteraceae* were enriched in HC (p < 0.001). Abbreviations: HC, healthy controls; MGD, meibomian gland dysfunction.

## Correlation between the relative abundance of *Bacteroides* and clinical characteristics

Further analysis was performed on the relative abundance of *Bacteroides* and clinical characteristics. We found a very weak positive correlation between increased relative abundance of *Bacteroides* and lid margin irregularities, lid margin vascularity, and displacement of MCJ (Beta coefficient = 0.015, 0.018, 0.015, p < 0.05), and a weak positive correlation between the increased relative abundance of *Bacteroides* and meibum quality score. (r = 0.32, p < 0.05). We also found a weak negative correlation between the increased relative abundance of *Bacteroides* and TBUT (r = -0.287, p < 0.05). However, there was no correlation between the relative abundance of *Bacteroides* and plugged MG orifices, OSDI, Schirmer test, and Oxford grading of ocular surface staining.

**Table 3. Tear cytokine levels of patients with MGD and healthy controls.**

| Parameter | MGD | MGD | | HCs | p-value (HCs vs. MGD) | p-value (ANOVA) |
|---|---|---|---|---|---|---|
| | | Severe | Moderate | | | |
| IL-1β (pg/mL; Mean ± SD) | 11.15 ± 9.58 | 12.24 ± 10.78 | 9.57 ± 7.53 | 10.03 ± 10.69 | 0.61 | 0.61 |
| IL-6 (pg/mL; Mean ± SD) | 290.63 ± 329.61 | 345.80 ± 389.42 | 210.87 ± 200.99 | 178.03 ± 324.33 | 0.12 | 0.13 |
| IL-17A (pg/mL; Mean ± SD) | 51.81 ± 34.33 | 55.68 ± 36.16 | 46.21 ± 31.68 | 33.68 ± 27.62 | **0.008** | **0.02**\* |

\* Post-hoc test using Tukey's test shows a significant difference between the severe MGD group and healthy controls (adjusted P-value = 0.0144)

Abbreviations: HC, healthy controls; MGD, meibomian gland dysfunction; IL, interleukin

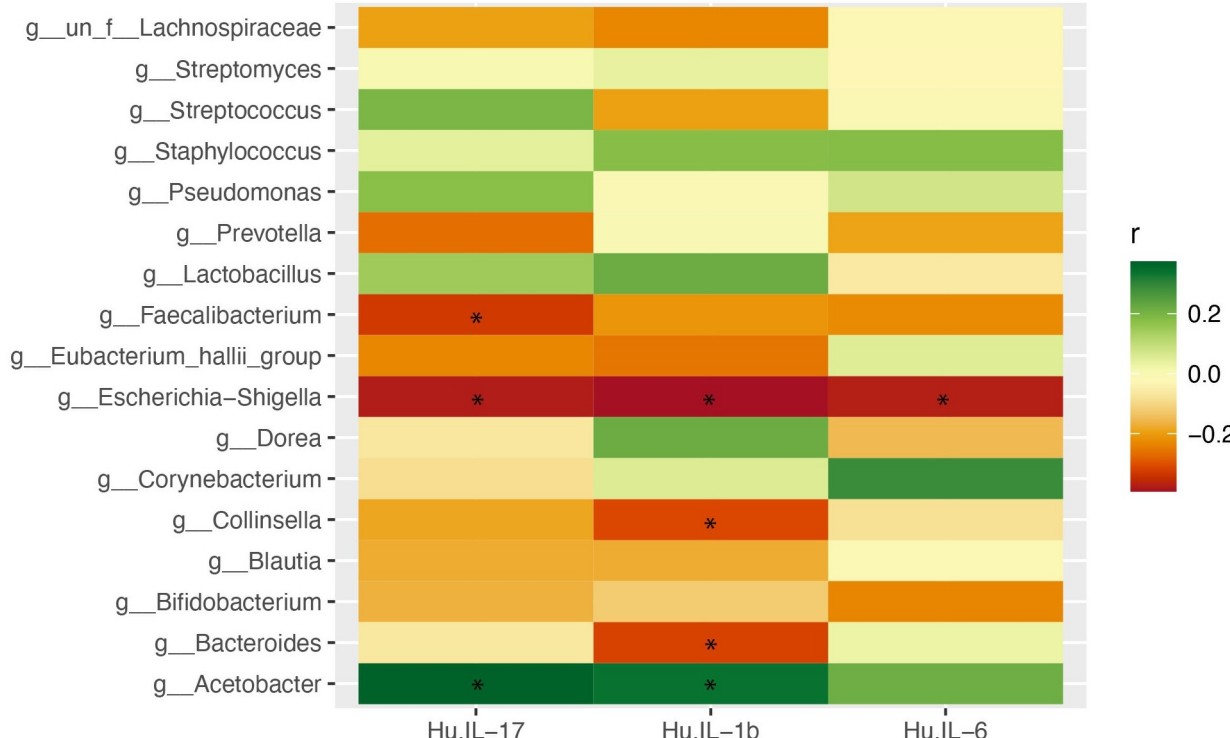

**Fig 4. This is the heatmap of correlations between core meibum microbiota and tear cytokines.** *p < 0.05 is considered statistically significant.

## Discussion and conclusions

To the best of our knowledge, few studies have identified meibum microbiota in patients with MGD using 16S rRNA gene sequencing, and this is the first study that analyzes the correlation between core meibum microbiota and tear cytokine levels in patients with MGD. Our study found no differences in the diversity of microbial communities between the MGD and HC groups. Nevertheless, Simpson's index indicated a markedly decreased α-diversity in patients with severe MGD than in those with moderate MGD, which suggest a relationship between reduced bacterial diversity and increased disease severity. However, the PCoA plot demonstrated that the samples from the MGD group were relatively well clustered compared with those from the HC group. In contrast to previous research based on shotgun metagenomic analysis [21], the community diversity was similar; however, distinct clusters were detected between the MGD and HC groups. These dissimilar results may be owing to different inclusion criteria, method of specimen collection, and sequencing techniques. Our study recruited patients with MGD without aqueous tear deficiency (ATD) while the previous study recruited patients with MGD regardless of ATD.

In this study, the predominant phyla in both MGD and HC groups were Firmicutes, Actinobacteria, Proteobacteria, and Bacteroidetes. In contrast, previous metagenomic sequencing studies have reported that the most predominant phyla were Proteobacteria, Actinobacteria, Firmicutes, and Bacteroidetes. In addition, a significantly decreased abundance of Proteobacteria was observed in the MGD group than in the HCs [21]. A previous study in Northwestern China reported that the predominant phyla in the eyelid margin and conjunctival sac of patients with blepharitis were Firmicutes, Proteobacteria, Bacteroidetes, and Actinobacteria.

In addition to the sites of collection and sequencing method, the geographic location also affected the composition of the microbiota [22, 23].

We reported *Blautia*, *Bacteroides*, *Escherichia-Shigella*, *Streptomyces*, *Corynebacterium*, *Bifidobacterium*, *Lactobacillus*, *Streptococcus*, *Faecalibacterium*, *Acetobacter*, *Prevotella*, *Lachnospiraceae*, *Dorea*, *Collinsella*, *Staphylococcus*, *Pseudomonas*, and *Eubacterium hallii* groups as the core meibum microbiota in both the MGD and HC groups. *Blautia*, *Bifidobacterium*, and *Faecalibacterium* were discovered in the meibum of patients with internal hordeolum treated with hypochlorous acid eyelid wipes [24]. These organisms are capable of producing butyrate, inhibiting nuclear factor kappa B signal transduction, and exerting anti-inflammatory effects that may play a significant role in maintaining equilibrium in the meibomian gland microenvironment [25].

In a healthy population, the most abundant microbiota in the eyelid skin and conjunctival sac were *Pseudomonas*, *Corynebacterium*, *Propionibacterium*, *Acinetobacter*, *Staphylococcus*, and *Streptococcus* [1, 26–28]. Only one published study has reported a healthy meibum microbiota using 16S rRNA gene sequencing, and the most abundant meibum microbiota were *Propionibacterium acnes* or *Pseudomonas sp*. [29]. However, the meibum was highly diverse in microbiota in young individuals, and this diversity decreased with increasing age. The aging population had an age range of 60–70 years, similar to our study groups. *Corynebacterium* sp. and *Neisseriaceae* dominate the meibum microbiota during aging. The conjunctival microbiota does not reflect the meibum microbiota in the aging population [29]. We postulated that meibum microbiota may represent the microenvironment of the meibomian gland better than other ocular surface microbiota.

Our study reported no difference in the abundances of *Propionibacterium*, *Streptococcus*, *Neisseriaceae*, *Staphylococcus*, *Pseudomonas*, and *Bacillus* between the MGD and HC groups (4.78% vs. 3.15%, 3.55% vs. 3.48%, 1.53% vs. 1.55%, 1.3% vs. 2.1%, 0.76 vs. 1.03%, and 0.15% vs. 0.29%, respectively; $p > 0.05$). *Corynebacterium* was more abundant in the HCs in our study; however, the difference was not statistically significant (7.59% vs. 5.14%; $p = 0.29$). Our results support previous evidence that *Corynebacterium* is abundant in the healthy conjunctival sac and meibum microbiota [1, 29, 30]. *Corynebacterium* acted as normal flora on the ocular surface because of its low pathogenicity [30] and may help prevent the overgrowth of other pathogenic bacteria.

Patients with MGD had a higher relative abundance of *Bacteroides* than HCs, particularly in the severe MGD group. Li *et al*. [31] reported that *Bacteroides* enriched the conjunctival microbiota in the dry eye. Nevertheless, the *Bacteroides* were the dominant microbiota in the non-MGD group, whereas the *Bacilli* were the dominant microbiota in the MGD group. However, they did not exclude patients with ATD from the study population. Our study compared patients with MGD without ATD and HCs without dry eye. We hypothesized that *Bacteroides* may associated with damaged lid margin structure (lid margin irregularities, lid margin vascularity, and displacement of MCJ), poor meibum quality, and shortened TBUT in patients with MGD. *Bacteroides* can penetrate the submucosal tissue and cause infection through the damaged mucosa [22]. Its virulence is due to its encapsulation and endotoxin-producing and highly antibiotic-resistant properties [32]. *Bacteroides* are scarcely found on the ocular surface and rarely detected using conventional culture methods [33]. The obstructed meibomian gland may alter the glandular environment, leading to an increased proportion of this microbe. There are few reports on *Bacteroides*-associated ocular infections, including blebitis, keratitis, and endophthalmitis after uneventful trabeculectomy [33] and postoperative endophthalmitis [34]. Due to a rare but severe infection from this potential pathogen, we suggest the diagnosis and treatment of MGD before conducting any ocular surgery.

*Novosphingobium*, although not the core meibum microbiota, was abundant in patients with MGD. Liang *et al*. [35] recently found a high relative abundance of *Novosphingobium* in the

conjunctival microbiota of patients with ocular *Demodex* infections (1.1% vs. 0.4%, p = 0.012). *Demodex* itself can destroy the meibomian and lacrimal glands, depleting the tear film's lipid and aqueous layers and resulting in DED. MGD may be a key factor in the alteration of ocular surface microbiota caused by *Demodex* infection. Sluch *et al.* [36] reported that *Acinetobacter*, *Cloacibacterium*, and *Novosphingobium* are strongly associated with keratitis caused by *Pseudomonas aeruginosa*. We analyzed the OTU number using differential abundance analysis, which confirmed that *Bacteroides* were enriched in the MGD group, whereas *Nocardia* and *Obscuribacteraceae* were enriched in the HC group. In contrast to a previous study, Zhao *et al.* also reported a high abundance of *Campylobacter coli*, *Campylobacter jejuni*, and *Enterococcus faecium* in the meibum of patients with MGD [21]. The difference in these findings may be caused by differences in the study population, specimen collection, and sequencing technique.

The common tear cytokines associated with MGD are IL-6, IL-17A, and IL-1β [6, 12, 13]. Expectedly, significantly higher IL-17A was detected in the MGD group (p = 0.008), particularly in the severe MGD group, than in the HC group. Moreover, IL-6 levels were increased in the MGD group; however, the difference was comparable. IL-17 and IL-6 promote Th17 function, which activates corneal epithelial barrier disruption and is thus associated with the pathogenesis of DED [37]. *Bacteroides* showed no correlation with IL-17A, despite being abundant in the MGD group, but showed a weak negative correlation with IL-1β. IL-1β is a proinflammatory cytokine that stimulates the antimicrobial immune response, inhibiting pathogen colonization, invasion, replication, and dissemination. Either decreased IL-1β causes microbial dysbiosis and increased abundance of *Bacteroides*, or, the increased abundance of *Bacteroides* causes the depletion of IL-1β, leading to microbial dysbiosis are needed to be resolved [38]. Further studies are required to evaluate the cause-and-effect relationship between the core meibum microbiota and tear cytokine levels as a primary outcome. Moreover, in addition to our included cytokines, other tear cytokines may also be considered.

Our study has some limitations. First, we did not obtain the microbiota from adjacent areas such as the lid margin and conjunctiva; therefore, their correlation with MGD could not be analyzed. Second, while disinfection of lid margin helps prevent contamination of meibum microbiota with lid margin microbiota, this process might have an effect on the resident bacteria in meibomian gland orifices. Third, the 16S rRNA gene sequencing method helps determine the bacterial community in the study population; however, it has some drawbacks as it is incapable of identifying non-bacterial microbiota, is not able to ensure the viability of the organism, and has weak phylogenetic ability at the species level [15]. Nevertheless, this method is more cost-effective than whole genome sequencing. Fourth, we did not focus on demodex infections so its relationship with MGD cannot be identified. Lastly, factors that may have led to different results included the study population, disease severity, geographic location, seasonal changes, swab location, depth, specimen collection method, and sequencing method, which should be considered in the future.

In conclusion, NGS analyses revealed decreased bacterial diversity in the meibum microbiota of patients with severe MGD. Significantly high abundance of *Bacteroides* and *Novosphingobium* were observed in the MGD group. The abundance of *Bacteroides* was negatively correlated with IL-1β levels in tears. However, the precise cause-effect relationship between the meibum microbiota and tear cytokine levels is yet to be elucidated.

## Supporting information

**S1 File. MGD severity grading scale.**
(DOCX)

## Acknowledgments

The authors sincerely appreciate the Cornea and Refractive Unit staff, Department of Ophthalmology, Faculty of Medicine, Chulalongkorn University, for providing patients' information and colleagues from the Department of Microbiology, Faculty of Medicine, Chulalongkorn University, for assistance with laboratory techniques.

## Author Contributions

**Conceptualization:** Ubonwan Rasaruck, Ngamjit Kasetsuwan, Thanachaporn Kittipibul, Tanittha Chatsuwan.

**Data curation:** Ubonwan Rasaruck, Pisut Pongchaikul.

**Formal analysis:** Ubonwan Rasaruck, Ngamjit Kasetsuwan, Thanachaporn Kittipibul, Pisut Pongchaikul, Tanittha Chatsuwan.

**Funding acquisition:** Ubonwan Rasaruck, Ngamjit Kasetsuwan.

**Investigation:** Ubonwan Rasaruck.

**Methodology:** Ubonwan Rasaruck, Ngamjit Kasetsuwan, Thanachaporn Kittipibul, Pisut Pongchaikul, Tanittha Chatsuwan.

**Project administration:** Ngamjit Kasetsuwan, Thanachaporn Kittipibul, Tanittha Chatsuwan.

**Resources:** Ubonwan Rasaruck, Ngamjit Kasetsuwan, Thanachaporn Kittipibul, Tanittha Chatsuwan.

**Software:** Pisut Pongchaikul.

**Supervision:** Ngamjit Kasetsuwan, Thanachaporn Kittipibul, Tanittha Chatsuwan.

**Validation:** Ubonwan Rasaruck, Ngamjit Kasetsuwan, Thanachaporn Kittipibul, Pisut Pongchaikul, Tanittha Chatsuwan.

**Visualization:** Ngamjit Kasetsuwan, Thanachaporn Kittipibul, Tanittha Chatsuwan.

**Writing – original draft:** Ubonwan Rasaruck.

**Writing – review & editing:** Ubonwan Rasaruck, Ngamjit Kasetsuwan, Thanachaporn Kittipibul, Pisut Pongchaikul, Tanittha Chatsuwan.

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
