## [Decision Letter · Decision Letter 0]

15 Aug 2023

PONE-D-23-22246Composition and diversity of meibum microbiota in meibomian gland dysfunction and the correlation with tear cytokine levelsPLOS ONE

Dear Dr. Kasetsuwan,

Thank you for submitting your manuscript to PLOS ONE. After careful consideration, we feel that it has merit but does not fully meet PLOS ONE’s publication criteria as it currently stands. Therefore, we invite you to submit a revised version of the manuscript that addresses the points raised during the review process.

We look forward to receiving your revised manuscript.

Kind regards,

Kofi Asiedu

Academic Editor

PLOS ONE

Journal Requirements:

Reviewers' comments:

Reviewer's Responses to Questions

**Comments to the Author**

1. Is the manuscript technically sound, and do the data support the conclusions?

Reviewer #1: Partly

Reviewer #2: Yes

Reviewer #3: Partly

2. Has the statistical analysis been performed appropriately and rigorously? 

Reviewer #1: Yes

Reviewer #2: Yes

Reviewer #3: Yes

3. Have the authors made all data underlying the findings in their manuscript fully available?

Reviewer #1: Yes

Reviewer #2: Yes

Reviewer #3: Yes

4. Is the manuscript presented in an intelligible fashion and written in standard English?

Reviewer #1: Yes

Reviewer #2: Yes

Reviewer #3: Yes

5. Review Comments to the Author

Reviewer #1: f both eyes met the inclusion criteria, the investigator randomly selected only one eye. There is insufficient basis for randomly selecting one eye.

Why there were no mild MGD patients included in the study?

There is insufficient reason to conclude that reduced bacterial diversity may be associated with increased MGD severity.

This MS did not obtain the microbiota from adjacent areas such as the lid margin and conjunctiva; therefore, their correlation with MGD is not clear.

The patients were enrolled from 6 November 2021 to 8 April 2022. The study spanned a large period of time, and did the seasons affect the flora of the meibomian glands?

Reviewer #2: Introduction:

Consider including a statement at the end of the introduction that clearly emphasizes the aim or objective of the study. This will help the readers understand the purpose of your research and what you intend to achieve through your investigation.

Methods:

Cytokine Number: State the exact number of cytokines that were investigated in your study. This information is important for readers to understand the scope of your cytokine analysis.

Test Order Collaboration: Provide information about the order in which tests were conducted or performed. Specify whether researchers collected meibum samples during the same visit when other clinical measurements were carried out.

DNA Measurement Before PCR: Clarify whether the authors measured the extracted DNA before performing PCR. This step is crucial to ensure the accuracy of subsequent PCR analyses.

Contaminant Filtering for Sequencing: Mention whether a contaminant filtering step was performed on the sequencing results. This step is important for ensuring the quality and reliability of microbiota data.

Disinfection of Lid Margin: While this process helps prevent contamination of meibum microbiota with lid margin microbiota, I am concerned that it might affect the resident bacteria in meibomian gland orifices, which can have a potential impact on MGD pathogenesis. Perhaps consider discussing this point as a potential limitation of the study.

Reviewer #3: PONE-D-23-22246

Composition and diversity of meibum microbiota in meibomian gland dysfunction and

the correlation with tear cytokine levels

The authors reported 16s rRNA sequencing of bacterial microbiota in patients with meibomian gland dysfunction (MGD) compared to healthy controls. They have excluded aqueous deficiency dry eye and included patients with MGD in two severity categories: severe and moderate. They also reported tear cytokine levels (IL-6, IL-17A, and IL-1β) and their correlation with microbiota. They found decreased alpha diversity in severe MGD patients compared to moderate MGD but no difference from healthy controls. Bacteroides genus was more abundant in MGD compared to healthy controls, and this was negatively correlated with the tear IL-1β levels. The study has significance. My comments are below.

Overall comments:

1. The authors reported their results in a precise manner; however, some important details may be added to their report. They should clarify how they define severe versus moderate MGD. The authors included the groups separately for some analyses and together for others, not in a systematic way. They should consider reporting their results comparing MGD vs. HC first and then subgroup analysis following that. Then, in the discussion, they should consider discussing their results in that respect.

2. Lid margin assessment was done according to table 1, but no analysis was noted regarding the impact of altered microbiome and/or cytokine level on lid margin structures or distinct findings such as vascularization or others. More detail is needed regarding meibum quality, expressibility, or any other features they assessed showed any significance regarding their analysis.

3. Discussion should include more comments from the authors. The authors repeat their results along with prior publications’ results, but rarely does a comment follow. At the end of each paragraph, the authors should include their comments.

4. Limitations should be expanded. The main disadvantage of 16s rRNA is not being able to characterize non-bacterial microbiome. It is particularly important in MGD; as we know, Demodex plays an important role. Demodex genes were not assessed, but the authors could have collected data on Demodex presence. This should be added to the limitations. Also, with NGS, viability is not assessed, which should be included in the limitation paragraph along with its implications.

Specific comments:

Page 3, line 49: Please consider replacing “inflamed” with “inflammatory.”

Page 3, line 54: Please specify where the “proliferation” occurs, including a reference. In the gland or lid margin or ocular surface?

Page 6, lines 112-113: Please include detail about the swab, what type of swab was used: cotton or flocked?

Page 8, line 172: Please consider rephrasing this sentence - unclear what they meant by 88 meibum and tear samples: 44 meibum and 44 tear samples?

Table 1: Please clarify what 3 groups were in the last column in the header; please add missing units for variables, clarify lid assessment variables: please clarify if it represents numbers of yes or something else, add % if categorical, and clarify what meibum quality represents. None of these features were explained in the methods. The authors should describe how they assessed these features listed here in the methods.

Page 10, lines 205-208: Novosphingobium is not seen in table 2 or figure 2, although reported to be more abundant along with Bacteroides in MGD. Please clarify.

Page 13, line 250: Did the authors mean correlated with Bacteroides “abundance rate”? Please revise.

Page 17, lines 343-346: A negative correlation between Bacteroides and IL1b deserves some highlight, and the authors should add some comments on what this might mean and what next steps should be taken.

Page 15, lines 300-301: Please include a reference.

6. PLOS authors have the option to publish the peer review history of their article (what does this mean?). If published, this will include your full peer review and any attached files.

Reviewer #1: **Yes: **Xiuming Jin

Reviewer #2: **Yes: **Azadeh Tavakoli

Reviewer #3: **Yes: **Sezen Karakus

---

## [Author Response · Author response to Decision Letter 0]

28 Sep 2023

Reviewers' comments:

Reviewer's Responses to Questions

Comments to the Author

1. Is the manuscript technically sound, and do the data support the conclusions?

Reviewer #1: Partly

Reviewer #2: Yes

Reviewer #3: Partly

2. Has the statistical analysis been performed appropriately and rigorously?

Reviewer #1: Yes

Reviewer #2: Yes

Reviewer #3: Yes

3. Have the authors made all data underlying the findings in their manuscript fully available?

Reviewer #1: Yes

Reviewer #2: Yes

Reviewer #3: Yes

4. Is the manuscript presented in an intelligible fashion and written in standard English?

Reviewer #1: Yes

Reviewer #2: Yes

Reviewer #3: Yes

5. Review Comments to the Author

Reviewer #1: If both eyes met the inclusion criteria, the investigator randomly selected only one eye. There is insufficient basis for randomly selecting one eye.

Meibomian gland dysfunction almost always affects both eyes. (1) We chose only one eye if both eyes met the inclusion criteria because we thought that the findings on the right eye would be more similar to the left eye in the same individual than in a different individual. 

Why there were no mild MGD patients included in the study?

We hypothesized that meibomian gland obstruction leads to dysbiosis of microbiota on the ocular surface. Bacterial overgrowth results in the production of inflammatory cytokines which cause subclinical inflammation(2). Several previous studies have observed increased cytokine levels in moderate to severe forms of dry eye (3-8) so our study group consisted of patients with moderate to severe MGD. 

We have added the reason in line 65-67.

There is insufficient reason to conclude that reduced bacterial diversity may be associated with increased MGD severity.

This MS did not obtain the microbiota from adjacent areas such as the lid margin and conjunctiva; therefore, their correlation with MGD is not clear.

We have removed the phrase in the conclusion that stated “However, reduced bacterial diversity may be associated with increased MGD severity.” (line 41-42)

We also clarified the limitations of our study “We did not obtain the microbiota from adjacent areas such as the lid margin and conjunctiva; therefore, their correlation with MGD could not be analyzed.” (line 425-427)

The patients were enrolled from 6 November 2021 to 8 April 2022. The study spanned a large period of time, and did the seasons affect the flora of the meibomian glands?

A total of 88 participants were enrolled in this study. Of these, 84 patients were enrolled from February to April 2022. Only 4 participants were enrolled in November to December 2021. The study was paused from around December 2021 to January 2022 due to COVID restrictions. However, we also mentioned the following as part of a description of the limitations of the study: “…factors that may have led to different results included the study population, disease severity, geographic location, seasonal changes, swab location, depth, specimen collection method, and sequencing method…” (line 433-435)

Reviewer #2: Introduction:

Consider including a statement at the end of the introduction that clearly emphasizes the aim or objective of the study. This will help the readers understand the purpose of your research and what you intend to achieve through your investigation.

We have added “In this study, we aimed to determine the meibum microbiota by NGS and its association with tear cytokine levels (IL-6, IL-17A, and IL-1β) in patients with MGD compared with healthy controls.” in the introduction. (line 72-74)

Methods:

Cytokine Number: State the exact number of cytokines that were investigated in your study. This information is important for readers to understand the scope of your cytokine analysis.

We have added the following line to the manuscript: “The specific tear cytokines, namely, IL-6, IL-17A, and IL-1β, were extracted from the Schirmer strips.” (line 144-145)

Test Order Collaboration: Provide information about the order in which tests were conducted or performed. Specify whether researchers collected meibum samples during the same visit when other clinical measurements were carried out.

We have clarified the methodology followed in our experiment as follows: “The participants were appointed for data collection which was performed in the following order in the same visit: (1) The baseline characteristics and OSDI questionnaire were recorded; (2) Schirmer test without anesthesia and tear sample collection were performed; (3) The ocular surface, lid, and meibomian gland assessments were performed by Ubonwan Rasaruck (U.R.) following the MGD severity grading scale, modified from Geerling et al. (9) (S1 File); (4) The meibum sample was collected by U.R..” (line 117-123)

DNA Measurement Before PCR: Clarify whether the authors measured the extracted DNA before performing PCR. This step is crucial to ensure the accuracy of subsequent PCR analyses.

We have added a description of this step in the manuscript “The concentration of extracted DNA was measured in ng/ul.” (line 143)

Contaminant Filtering for Sequencing: Mention whether a contaminant filtering step was performed on the sequencing results. This step is important for ensuring the quality and reliability of microbiota data.

We have added a description of this step in the manuscript “The contaminant filtering step was performed by using positive and negative controls and filtering out rare amplicon sequence variants.” (line 157-159)

Disinfection of Lid Margin: While this process helps prevent contamination of meibum microbiota with lid margin microbiota, I am concerned that it might affect the resident bacteria in meibomian gland orifices, which can have a potential impact on MGD pathogenesis. Perhaps consider discussing this point as a potential limitation of the study.

In this step, the eyelid margin was carefully sterilized while avoiding the area of the meibomian gland orifices. We have added a description of this on line 132-134: “The disinfection of the eyelid margin was carefully done while avoiding the area of meibomian gland orifices.”

Reviewer #3: PONE-D-23-22246

Composition and diversity of meibum microbiota in meibomian gland dysfunction and

the correlation with tear cytokine levels

The authors reported 16s rRNA sequencing of bacterial microbiota in patients with meibomian gland dysfunction (MGD) compared to healthy controls. They have excluded aqueous deficiency dry eye and included patients with MGD in two severity categories: severe and moderate. They also reported tear cytokine levels (IL-6, IL-17A, and IL-1β) and their correlation with microbiota. They found decreased alpha diversity in severe MGD patients compared to moderate MGD but no difference from healthy controls. Bacteroides genus was more abundant in MGD compared to healthy controls, and this was negatively correlated with the tear IL-1β levels. The study has significance. My comments are below.

Overall comments:

1. The authors reported their results in a precise manner; however, some important details may be added to their report. They should clarify how they define severe versus moderate MGD. 

We clarified in line 95-106

“The patients were diagnosed as having moderate MGD if they had at least one of the following criteria: (1) OSDI score 23–32; (2) lid margin vascularity or plugged meibomian orifice; (3) meibum quality score 8–12; (4) meibum expressibility grade 2; (5) Oxford grading score 4–10. The patients were diagnosed as having severe MGD if they had at least one of the following criteria: (1) OSDI score 33–100; (2) displacement of the mucocutaneous junction; (3) meibum quality score ≥ 13; (4) meibum expressibility grade 3; (5) Oxford grading score 11–15. Meibum quality is assessed at eight glands in the central third of the lower eyelid; grade 0 = clear, grade 1 = cloudy, grade 2 = cloudy with granular debris, grade 3 = thick, like toothpaste. Meibum expressibility is assessed at five glands in central third of lower eyelid; grade 0 = all, grade 1 = 3–4, grade 2 = 1–2, grade 3 = 0, modified from Geerling et al. (9) (S1 File).”

The authors included the groups separately for some analyses and together for others, not in a systematic way. They should consider reporting their results comparing MGD vs. HC first and then subgroup analysis following that.

We have rearranged the results section in a more systematic manner. (line 200-312).

Then, in the discussion, they should consider discussing their results in that respect.

We have rearranged the discussion section appropriately (line 314-442).

2. Lid margin assessment was done according to table 1, but no analysis was noted regarding the impact of altered microbiome and/or cytokine level on lid margin structures or distinct findings such as vascularization or others. More detail is needed regarding meibum quality, expressibility, or any other features they assessed showed any significance regarding their analysis.

Further analysis was performed on the relative abundance of Bacteroides and clinical characteristics. We found a very weak positive correlation between increased relative abundance of Bacteroides and lid margin irregularities, lid margin vascularity, and displacement of MCJ. (Beta coefficient = 0.015, 0.018, 0.015, p < 0.05) and weak positive correlation between increased relative abundance of Bacteroide and meibum quality score. (r = 0.32, p < 0.05). We also found a weak negative correlation between the increased relative abundance of Bacteroides and TBUT (r = -0.287, p < 0.05). However, there was no correlation between the relative abundance of Bacteroides and plugged MG orifices, OSDI, Schirmer test, and Oxford grading of ocular surface staining. (line 301-312)

We hypothesized that Bacteroides may associated with damaged lid margin structure (lid margin irregularities, lid margin vascularity, and displacement of MCJ), poor meibum quality, and shortened TBUT in patients with MGD. Bacteroides can penetrate the submucosal tissue and cause infection through the damaged mucosa (10). Its virulence is due to its encapsulation and endotoxin-producing and highly antibiotic-resistant properties (11). (line 382-387)

3. Discussion should include more comments from the authors. The authors repeat their results along with prior publications’ results, but rarely does a comment follow. At the end of each paragraph, the authors should include their comments.

We have rearranged the discussion section appropriately (line 314-442).

4. Limitations should be expanded. The main disadvantage of 16s rRNA is not being able to characterize non-bacterial microbiome. It is particularly important in MGD; as we know, Demodex plays an important role. Demodex genes were not assessed, but the authors could have collected data on Demodex presence. This should be added to the limitations. Also, with NGS, viability is not assessed, which should be included in the limitation paragraph along with its implications.

We have added the following on line 427-433 

“Secondly, the 16S rRNA gene sequencing method helps determine the bacterial community in the study population; however, it has some drawbacks as it is incapable of identifying non-bacterial microbiota, is not able to ensure the viability of the organism, and has weak phylogenetic ability at the species level (12). Nevertheless, this method is more cost-effective than whole genome sequencing. Third, we did not focus on demodex infestration so its relationship with MGD can not be identified.”

Specific comments:

Page 3, line 49: Please consider replacing “inflamed” with “inflammatory.”

We have made corrections on line 49 as follows: “Inflammatory obstructive MGD highly correlates with hordeolum, conjunctivitis, and keratitis”.

Page 3, line 54: Please specify where the “proliferation” occurs, including a reference. In the gland or lid margin or ocular surface?

We have made corrections as follows on line 53-55: “One of the key pathophysiologies of MGD is hyperviscous meibum, which leads to meibum stasis and eventually results in the proliferation of pathogenic bacteria on the ocular surface as well as inside the meibomian gland(2).”

Page 6, lines 112-113: Please include detail about the swab, what type of swab was used: cotton or flocked?

We have made corrections as follows on line 134-135: “The meibum was squeezed using a meibomian gland compressor and collected with a dry sterile cotton swab.”

Page 8, line 172: Please consider rephrasing this sentence - unclear what they meant by 88 meibum and tear samples: 44 meibum and 44 tear samples?

We have made corrections as follows on line 202 “We collected 88 meibum samples and 88 tear samples…”

Table 1: Please clarify what 3 groups were in the last column in the header; please add missing units for variables, clarify lid assessment variables: please clarify if it represents numbers of yes or something else, add % if categorical, and clarify what meibum quality represents. None of these features were explained in the methods. The authors should describe how they assessed these features listed here in the methods.

We have made corrections to Table 1 and the methods section on line 102-106 (modified from (9)(Supplementary file 1.))

Page 10, lines 205-208: Novosphingobium is not seen in table 2 or figure 2, although reported to be more abundant along with Bacteroides in MGD. Please clarify.

We have included the following clarification on line 246-248 “Novosphingobium, although not among the ten most predominant microbiota, had significantly higher relative abundances in the MGD group, compared with the HCs. (0.14% vs. 0.004%, p = 0.012)”

Page 13, line 250: Did the authors mean correlated with Bacteroides “abundance rate”? Please revise.

We have made the following corrections on line 288-289: “Although there were significantly higher levels of IL-17A in MGD, IL-17A was not correlated with the relative abundance of Bacteroides” 

Page 17, lines 343-346: A negative correlation between Bacteroides and IL1b deserves some highlight, and the authors should add some comments on what this might mean and what next steps should be taken.

We have added the following on line 417-421: “IL-1β is a proinflammatory cytokine that stimulates the antimicrobial immune response, inhibiting pathogen colonization, invasion, replication, and dissemination. Either decreased IL-1β causes microbial dysbiosis and increased abundance of Bacteroides, or, increased abundance of Bacteroides may cause the depletion of IL-1β, leading to microbial dysbiosis are needed to be resolved.(13)”

Page 15, lines 300-301: Please include a reference.

We have added a relevant reference on line 359.

6. PLOS authors have the option to publish the peer review history of their article (what does this mean?). If published, this will include your full peer review and any attached files.

Do you want your identity to be public for this peer review? For information about this choice, including consent withdrawal, please see our Privacy Policy.

Reviewer #1: Yes: Xiuming Jin

Reviewer #2: Yes: Azadeh Tavakoli

Reviewer #3: Yes: Sezen Karakus

1. Reza A. Badian TPU, Xiangjun Chen, Øygunn Aass Utheim, Sten Ræder, Ann Elisabeth Ystenæs, Bente Monica Aakre, Vibeke Sundling. Meibomian gland dysfunction is highly prevalent among first‐time visitors at a Norwegian dry eye specialist clinic. Scientific Reports. 2021;11.

2. Knop E, Knop N, Millar T, Obata H, Sullivan DA. The international workshop on meibomian gland dysfunction: report of the subcommittee on anatomy, physiology, and pathophysiology of the meibomian gland. Invest Ophthalmol Vis Sci. 2011;52(4):1938-78.

3. Pflugfelder SC, Jones D, Ji Z, Afonso A, Monroy D. Altered cytokine balance in the tear fluid and conjunctiva of patients with Sjogren's syndrome keratoconjunctivitis sicca. Curr Eye Res. 1999;19(3):201-11.

4. Solomon A, Dursun D, Liu Z, Xie Y, Macri A, Pflugfelder SC. Pro- and anti-inflammatory forms of interleukin-1 in the tear fluid and conjunctiva of patients with dry-eye disease. Invest Ophthalmol Vis Sci. 2001;42(10):2283-92.

5. Tishler M, Yaron I, Geyer O, Shirazi I, Naftaliev E, Yaron M. Elevated tear interleukin-6 levels in patients with Sjogren syndrome. Ophthalmology. 1998;105(12):2327-9.

6. Massingale ML, Li X, Vallabhajosyula M, Chen D, Wei Y, Asbell PA. Analysis of inflammatory cytokines in the tears of dry eye patients. Cornea. 2009;28(9):1023-7.

7. De Paiva CS, Chotikavanich S, Pangelinan SB, Pitcher JD, 3rd, Fang B, Zheng X, et al. IL-17 disrupts corneal barrier following desiccating stress. Mucosal Immunol. 2009;2(3):243-53.

8. Enriquez-de-Salamanca A, Castellanos E, Stern ME, Fernandez I, Carreno E, Garcia-Vazquez C, et al. Tear cytokine and chemokine analysis and clinical correlations in evaporative-type dry eye disease. Mol Vis. 2010;16:862-73.

9. Geerling G, Tauber J, Baudouin C, Goto E, Matsumoto Y, O'Brien T, et al. The international workshop on meibomian gland dysfunction: report of the subcommittee on management and treatment of meibomian gland dysfunction. Invest Ophthalmol Vis Sci. 2011;52(4):2050-64.

10. Wang C, Dou X, Li J, Wu J, Cheng Y, An N. Composition and Diversity of the Ocular Surface Microbiota in Patients With Blepharitis in Northwestern China. Front Med (Lausanne). 2021;8:768849.

11. Mobile genetic elements in the genus Bacteroides, and their mechanism(s) of dissemination. Mai Nguyen, Gayatri Vedantam 2011;1(3):187-96.

12. Janda JM, Abbott SL. 16S rRNA gene sequencing for bacterial identification in the diagnostic laboratory: pluses, perils, and pitfalls. J Clin Microbiol. 2007;45(9):2761-4.

13. LaRock CN TJ, LaRock DL, Olson J, O'Donoghue AJ, Robertson AA, Cooper MA, Hoffman HM, Nizet V. IL-1β is an innate immune sensor of microbial proteolysis. Science Immunology. 2016;1(2).

---

## [Decision Letter · Decision Letter 1]

7 Nov 2023

PONE-D-23-22246R1Composition and diversity of meibum microbiota in meibomian gland dysfunction and the correlation with tear cytokine levelsPLOS ONE

Dear Dr. Kasetsuwan,

Thank you for submitting your manuscript to PLOS ONE. After careful consideration, we feel that it has merit but does not fully meet PLOS ONE’s publication criteria as it currently stands. Therefore, we invite you to submit a revised version of the manuscript that addresses the points raised during the review process.

We look forward to receiving your revised manuscript.

Kind regards,

Kofi Asiedu

Academic Editor

PLOS ONE

Journal Requirements:

Reviewers' comments:

Reviewer's Responses to Questions

**Comments to the Author**

1. If the authors have adequately addressed your comments raised in a previous round of review and you feel that this manuscript is now acceptable for publication, you may indicate that here to bypass the “Comments to the Author” section, enter your conflict of interest statement in the “Confidential to Editor” section, and submit your "Accept" recommendation.

Reviewer #1: (No Response)

Reviewer #3: All comments have been addressed

2. Is the manuscript technically sound, and do the data support the conclusions?

Reviewer #1: (No Response)

Reviewer #3: Yes

3. Has the statistical analysis been performed appropriately and rigorously? 

Reviewer #1: (No Response)

Reviewer #3: Yes

4. Have the authors made all data underlying the findings in their manuscript fully available?

Reviewer #1: (No Response)

Reviewer #3: Yes

5. Is the manuscript presented in an intelligible fashion and written in standard English?

Reviewer #1: (No Response)

Reviewer #3: Yes

6. Review Comments to the Author

Reviewer #1: I have no additional comments for the authors. This manuscript had been revised according my comments ,and can be accept.

Reviewer #3: Page 6, lines 133-134: Please clarify how the meibomian gland orifices can be avoided when disinfecting eyelid margins. It does not sound that it can be promised. I suggest removing this sentence but adding the Reviewer’s comment in the limitation.

Table 1. Please revise the following points:

- Spell out MGD in the table title

- Correct the misspelled word “particapants”,

- right and left eyes: please exchange (eyes) with (number).

- Please remove (number of yes) after lid assessment and add (present, number) next to each parameter

Page 11, line 221-223: Please remove “for the severe MGD, moderate MGD, and HC groups” as it is redundant and confusing. Clarify what is “no different from the HCs”. Decreased diversity in severe MGD than in HCs? Revise accordingly.

7. PLOS authors have the option to publish the peer review history of their article (what does this mean?). If published, this will include your full peer review and any attached files.

Reviewer #1: **Yes: **Xiuming Jin

Reviewer #3: **Yes: **Sezen Karakus

---

## [Author Response · Author response to Decision Letter 1]

24 Nov 2023

Reviewers' comments:

Reviewer's Responses to Questions

Comments to the Author

1. If the authors have adequately addressed your comments raised in a previous round of review and you feel that this manuscript is now acceptable for publication, you may indicate that here to bypass the “Comments to the Author” section, enter your conflict of interest statement in the “Confidential to Editor” section, and submit your "Accept" recommendation.

Reviewer #1: (No Response)

Reviewer #3: All comments have been addressed

2. Is the manuscript technically sound, and do the data support the conclusions?

Reviewer #1: (No Response)

Reviewer #3: Yes

3. Has the statistical analysis been performed appropriately and rigorously?

Reviewer #1: (No Response)

Reviewer #3: Yes

4. Have the authors made all data underlying the findings in their manuscript fully available?

Reviewer #1: (No Response)

Reviewer #3: Yes

5. Is the manuscript presented in an intelligible fashion and written in standard English?

Reviewer #1: (No Response)

Reviewer #3: Yes

6. Review Comments to the Author

Reviewer #1: I have no additional comments for the authors. This manuscript had been revised according my comments ,and can be accept.

Reviewer #3: Page 6, lines 133-134: Please clarify how the meibomian gland orifices can be avoided when disinfecting eyelid margins. It does not sound that it can be promised. I suggest removing this sentence but adding the Reviewer’s comment in the limitation.

We have removed the sentence “The disinfection of the eyelid margin was carefully done while avoiding the area of meibomian gland orifices.” (page 6, line 124-126)

We also add the limitations of our study “Second, while disinfection of lid margin helps prevent contamination of meibum microbiota with lid margin microbiota, this process might have an effect on the resident bacteria in meibomian gland orifices.” (page 21, line 418-420)

Table 1. Please revise the following points:

- Spell out MGD in the table title

- Correct the misspelled word “particapants”,

- right and left eyes: please exchange (eyes) with (number).

- Please remove (number of yes) after lid assessment and add (present, number) next to each parameter

We have made correction in table 1., page 9-10)

Page 11, line 221-223: Please remove “for the severe MGD, moderate MGD, and HC groups” as it is redundant and confusing. Clarify what is “no different from the HCs”. Decreased diversity in severe MGD than in HCs? Revise accordingly.

We have removed the phase “for the severe MGD, moderate MGD, and HC groups” (page 11, line 212-213)

We also revised the sentence “Simpson’s index showed significantly decreased α-diversity in patients with severe MGD than those with moderate MGD. (p = 0.045; Fig 1). However, there was no difference in α-diversity between patients with severe MGD and HCs and patients with moderate MGD and HCs (p > 0.05; Fig 1).” (page 11, line 213-217)

7. PLOS authors have the option to publish the peer review history of their article (what does this mean?). If published, this will include your full peer review and any attached files.

Do you want your identity to be public for this peer review? For information about this choice, including consent withdrawal, please see our Privacy Policy.

Reviewer #1: Yes: Xiuming Jin

Reviewer #3: Yes: Sezen Karakus

---

## [Editor Report · Decision Letter 2]

10 Dec 2023

Composition and diversity of meibum microbiota in meibomian gland dysfunction and the correlation with tear cytokine levels

PONE-D-23-22246R2

Dear Dr. Kasetsuwan,

We’re pleased to inform you that your manuscript has been judged scientifically suitable for publication and will be formally accepted for publication once it meets all outstanding technical requirements.

Kind regards,

Kofi Asiedu

Academic Editor

PLOS ONE
---

## [Editor Report · Acceptance letter]

14 Dec 2023

PONE-D-23-22246R2 

PLOS ONE

Dear Dr. Kasetsuwan, 

I'm pleased to inform you that your manuscript has been deemed suitable for publication in PLOS ONE. Congratulations! Your manuscript is now being handed over to our production team.

Kind regards, 

on behalf of

Dr. Kofi Asiedu 

Academic Editor

PLOS ONE